# Digital Device Use, Risk of Cognitive Impairment, and Cognition in Healthy Older Adults: The Role of Cognitive Reserve

**DOI:** 10.3390/healthcare11212822

**Published:** 2023-10-25

**Authors:** Carolyn Liang, Ponnusamy Subramaniam, Nurul Syasya Mohd Ridzwan Goh, Tay Kok Wai, Ahmed A. Moustafa

**Affiliations:** 1Clinical Psychology Programme, Centre of Health Care Sciences, Faculty of Health Sciences, Universiti Kebangsaan Malaysia, Jln Raja Muda Abdul Aziz, Kuala Lumpur 50300, Malaysia; liangcarolyn.s@gmail.com (C.L.); syasyagoh@ukm.edu.my (N.S.M.R.G.); 2Centre for Health Ageing & Wellness (HCARE), Faculty of Health Sciences, Universiti Kebangsaan Malaysia, Jln Raja Muda Abdul Aziz, Kuala Lumpur 50300, Malaysia; taykw@utar.edu.my; 3Department of Psychology and Counseling, Faculty of Arts and Social Science, Tunku Abdul Rahman University, Kampar 31900, Perak, Malaysia; 4Center for Data Analytics, School of Psychology, Faculty of Society and Design, Bond University, Gold Coast, QLD 4226, Australia; amoustaf@bond.edu.au

**Keywords:** digital device, cognitive reserve, cognition, cognitive impairment, older adults, cognitive function

## Abstract

Neuroprotective factors are essential to successful ageing. As such, digital device use was proposed as an easily accessible and stimulating available cognitive activity to enhance brain function. Nonetheless, there was a lack of studies inspecting the connection between digital device use and cognitive reserve, the risk of cognitive impairment, and cognition. This study aims to investigate the potential mediator and moderator of the association between digital device use, cognitive reserve, the risk of cognitive impairment, and cognition among healthy older adults. A quantitative cross-sectional study was conducted to investigate the relationship between digital device use and cognitive reserve, the risk of cognitive impairment, and cognition. A total of 210 healthy older adults were recruited through purposive sampling. The results obtained from this study revealed that there was a significant difference in cognitive reserve and cognition between healthy older adults who use a digital device for communication purpose only and who use a digital device for multiple purposes. A significant relationship was also found between cognitive reserve, the risk of cognitive impairment, and cognition. Although digital device use was found to be significantly associated with cognitive reserve and cognition, it was not significantly associated with the risk of cognitive impairment. Cognitive reserve partially mediated the relationship between digital device use and cognition, supporting the notion that cognitive reserve acts as an underlying mechanism in the relationship between digital device use and cognition. Hence, digital device use was suggested to be a good daily intervention for healthy older adults to build on their cognitive reserve and potentially protect their cognition from declining. Nevertheless, relying on digital device use alone is not sufficient, and other activities should be explored to enhance cognitive reserve among healthy older adults.

## 1. Introduction

The health issues of the ageing population are a worldwide concern, with the number of people aged 60 years old and above growing to approximately 2 billion in the year 2050 [1]. It was estimated that the population aged 65 years old and above will reach 14.5% in 2040 in Malaysia [2]. The ageing population has an impact on health care, economic, and social issues [3]. Hence, it is important for the nation to prepare for the ageing population to be physically and cognitively active and healthy. Inevitably, the current perspective on ageing is mostly negative, and there are increased efforts to change the negative aspects of ageing and combat the negative stigma related to ageing [4]. A positive view about ageing can be approached in terms of successful ageing [5], which emphasizes healthy cognition or the brain. The concept of successful ageing includes different areas of life, such as circumventing disease and disability, maintaining functioning cognition, mental and physical health, being active in life, and being psychologically adaptive during old age [6]. Important factors associated with successful ageing are neuroprotective factors [7].

Older adults in Malaysia have high accessibility to digital devices as compared with other older adults in developed countries [8]. Older adults in Malaysia mainly use mobile phones, smartphones, laptops, and tablets to assess the Internet [9]. There are reports on the benefits of digital device use among older adults including delaying cognitive deterioration, improving connectivity with their social circles, maintaining basic daily activities, and supporting health care services [10,11,12]. The pandemic has brought a new perspective on the use of digital devices as people depend more on digital devices [11].

As the population is ageing, many older adults develop a risk for cognitive impairment, and some experience greater cognitive deterioration towards MCI and Alzheimer’s diseases. In Malaysia, the prevalence of dementia is 8.5% for older adults aged 60 years old and above [13]. Cognitive Reserve Theory proposed by Stern [14] asserts that some older adults can delay the onset of cognitive impairment despite the ageing process as they are able to retain better cognitive function. Engagement in mental and physical activity contributes to higher cognitive reserve in older adults [15]. A systematic review reported that a modifiable lifestyle activity can affect an individual’s cognitive reserve [16]. These modifiable lifestyle activities include physical and cognitive leisure activities. In fact, cognitive activities can be the sole predictor in some of the studies in contributing to cognitive reserve [16]. Meanwhile, the development of cognitive impairment is not immediate and acute, but is mostly seen as a gradual process. Individuals differ in their level of risk of developing cognitive impairment. Interestingly, cognitive leisure activities were found to be associated with a decrease in the risk of cognitive impairment [17]. Hence, intervention such as lifestyle modification towards cognitively stimulating activities among healthy older adults is important to prevent cognitive impairment.

As such, identifying neuroprotective factors is crucial to promote a healthy brain and protect against cognitive decline. One such possible neuroprotective factor for cognitive decline is digital device use in providing cognitively challenging and stimulating activities. The digital device is readily available and increasingly accessible in today’s era. Older people owning digital devices such as mobile phones are becoming common, with the majority of older adults in Malaysia using mobile phones [18]. A previous study posited that digital device use is related to cognitive function among older adults [19]. In Malaysia, a web-based activity was developed for older adults to reduce the risk of mild cognitive impairment [20]. However, there are limited studies researching digital device use and the mechanism of cognitive reserve, which involves the risk of cognitive impairment and cognition.

Upon the development of technology, researchers tried to use digital devices to deliver an intervention to improve older adults’ cognition [21]. A longitudinal study among older adults suggested that digital devices can be effective in delaying cognitive deterioration even without an intervention designed by professionals to specifically improve cognition [10]. On the other hand, it is important to take note that not all older adults who own a digital device use it with the same engagement level or purpose. Previous research divides older adults into daily users and non-daily users to investigate the impact of digital device use [19]. However, this left out important information such as the purpose of use, which will potentially lead to different levels of cognitive stimulation. Investigating the length of use is not a reliable indicator because older adults mainly utilize digital devices for communicating and connecting with others and less for other purposes, such as games and activities that require more cognitive effort [22]. Thus, this study provides important insight into how various uses of digital devices impact cognition in older adults differently.

In addition to that, Jin et al. [10] posited that the exact mechanism of how digital device use protects against cognitive decline is not clear from a theoretical perspective. It is speculated that digital devices build on the cognitive reserve that is suppressed against the manifestation of cognitive impairment. This provides an opportunity to confirm how cognitive reserve is related to the risk of cognitive impairment and cognition in healthy adults. In older adults with normal cognition, stronger cognitive reserve was found to be associated with a higher level of cognition [23]. However, in older adults who develop Mild Cognitive Impairment (MCI), there is a more rapid cognitive decline discovered in older adults without stronger cognitive reserve. This finding suggested that cognitive reserve mediates the relationship between cognitive pathology and cognition. As such, this model requires further verification. Most importantly, there is a lack of studies on how digital devices are associated with cognitive reserve, the risk of cognitive impairment, and cognition.

This study focused on three important constructs, cognitive reserve, the risk of cognitive impairment, and cognition. The relationship between these three constructs will be investigated based on the working model posited by Franzmeier et al. [24]. Based on this working model, this research proposes that cognition, the risk of cognitive impairment, and cognitive reserve are three different constructs. This working model potentially suggests a direct effect pathology towards cognition, and cognitive reserve acts as a latent construct that moderates the effect of MCI on cognition, which is promoted by different types of protective factors. A previous study suggested that cognitive reserve acts as a mediator in the effects of modifiable lifestyle factors, which are composed of social, physical, and cognitive activities towards cognition [25]. The mediation effect of cognitive reserve was found to be involved in 21% of the overall effect. Previous research also proposed that digital device use is a significant cognitive stimulation activity that explains the variation in cognitive reserve [26]. Other than that mediating effect, a previous study also posited that cognitive reserve has a moderation effect on cognition [27]. Thus, this research provides an opportunity to investigate the role of cognitive reserve acting as a moderator and mediator.

In addition, this research proposed to measure the construct of pathology through assessing the risk of cognitive impairment among healthy older adults. This is because pathology among healthy older adults cannot be conceptualised as a direct measurement of cognitive impairment through biological or clinical manifestation, as the pathology will not be evident before it develops. Furthermore, this research is interested in studying the prevention of pathology through reducing the risk of cognitive impairment. The individual difference in the risk of cognitive impairment is measurable in healthy older adults through the measurement of risk factors. In this regard, cognitively stimulating activities were found to reduce the risk of cognitive impairment [17]. As such, it is predicted that digital device use, a type of cognitive stimulating activity, will also contribute towards cognitive reserve, which mediates the effect of the risk of cognitive impairment on cognition. However, the association between digital device use with cognitive reserve, the risk of cognitive impairment, and cognition was not clear. This research proposed that digital device use among healthy older adults could act as a protective factor towards these three constructs, cognitive reserve, the risk of cognitive impairment, and cognition. Therefore, the following hypotheses are proposed and tested:Healthy older adults who use digital devices that require more cognitive effort will have statistically higher cognitive reserve and cognition and a lower risk of cognitive impairment compared to healthy older adults who use digital devices only for communicative purposes.There will be a significant correlation between cognitive reserve with the risk of cognitive impairment and cognition.Digital device use will be significantly associated with cognitive reserve, the risk of cognitive impairment, and cognition among healthy older adults.There is mediating and moderating effect of cognitive reserve in the relationship between digital device use with the risk of cognitive impairment and cognition.

Hence, the main objective of this research was to investigate the potential mediators and moderators of the association between digital device use, cognitive reserve, the risk of cognitive impairment, and cognition among healthy older adults.

## 2. Materials and Methods

### 2.1. Participants

A quantitative cross-sectional study was conducted with older adults aged 60 years old and above. A total of 210 participants who fulfilled the inclusion and exclusion criteria were recruited from a community around the Klang Valley area between the period of August 2021 and June 2022. The inclusion criteria were as follows: (i) healthy older adults aged 60 years old and above who (ii) have the ability to speak and understand Malay or English and (iii) are capable of understanding and providing informed consent. The exclusion criteria were as follows: (i) not using any digital devices; (ii) cognitive or psychiatric problems that impede informed consent; (iii) serious chronic illness or cognitive impairment, including dementia, heart disease, heart failure, stroke, chronic lung disease and cancer, major psychiatric illness, and other unspecified serious chronic illnesses that jeopardise scores on the Malay Mini-Mental Status Examination (M-MMSE) developed by Folstein et al. [28]; (iv) uncorrected hearing and visual impairment; (v) an M-MMSE score of 23 points and below, which indicates cognitive impairment or signs of dementia, even where there is no official diagnosis from a registered medical doctor; and (vi) a score of 5 points or above on the Malay Geriatric Depression Scale (M-GDS-14) developed by Teh and Che Ismail [29] which indicates signs of depressive symptoms.

### 2.2. Screening Tools and Study Variables

Initially, all relevant sociodemographic information was collected from participants. This was followed by the M-MMSE [28] and M-GDS-14 [29] to screen participants’ eligibility according to inclusion and exclusion criteria. Four standard and validated instruments were administered by the researcher via face-to-face interviews with participants to collect information regarding digital device use, cognitive reserve, the risk of cognitive impairment, and cognition. A detailed description of each research tool is provided in the following subsections.

#### 2.2.1. Malay Mini-Mental State Examination (M-MMSE)

The M-MMSE is an 11-item questionnaire [28] that aims to measure cognitive function in five areas, which are orientation, registration, attention and calculation, recall, and language. The test–retest reliability was reported at 0.45 to 0.50 with a one-year interval among healthy samples [30]. On the other hand, convergent validity was reported to be significantly correlated to verbal tasks. For the Malay version of the MMSE (M-MMSE), the Cronbach’s alpha reported was 0.70, and the validity of diagnosing dementia was also tested to be satisfactory [31].

#### 2.2.2. Geriatric Depression Scale (M-GDS-14)

The Malay version of the M-GDS-14 was modified and translated [29]. There are a total of 14 items to screen depressive symptoms, and the M-GDS-14 was designed to be self-administered. The M-GDS-14 was reported to have a Cronbach’s alpha of 0.84 and a test–retest reliability of 0.84. There is also good concurrent validity compared with the Montgomery–Åsberg Depression Rating Scale 18.

#### 2.2.3. Malay Cognitive Reserve Scale (M-CRS)

The CRS is a questionnaire measuring cognitive reserve developed by Leon et al. [26]. There are a total of 24 questions measuring participation in cognitively stimulating activities throughout the individual’s lifetime. This questionnaire was translated into the Malay version of the M-CRS using the back-translation method, and a validation study was conducted. The test has high reliability, with a Cronbach’s alpha of 0.78. The validity of the M-CRS was proven via correlation with years of education, occupational attainment, the M-MMSE, and Addenbrooke’s Cognitive Examination-III (ACE-III).

#### 2.2.4. TUA-WELLNESS

TUA-WELLNESS was developed by Vanoh et al. [32] as a screening tool to screen for mild cognitive impairment among older adults. There are a total of 10 items covering major risk factors of mild cognitive impairment. Item 7 has two different questions for men and women. Item 8 also has two different questions for Muslims and non-Muslims. A lower score indicates a higher risk for MCI and a higher score indicates a lower risk for MCI. The cut-off points for older adults who are at risk of MCI is 11, with an AUC value of 0.84, which has good discriminating power.

#### 2.2.5. Addenbrooke’s Cognitive Examination-III (ACE-III)

ACE-III was developed for the purpose of cognitive assessment in five major domains: attention, memory, fluency, language, and visuospatial components [33]. In particular, Hsieh et al. [33] reported good validity in diagnosing dementia impairment using ACE-III. ACE-III had also been translated into Malay and reported to have good reliability, with a Cronbach’s alpha coefficient of 0.83 and an intraclass correlation coefficient of 0.96 [34]. In addition, the Malay version of ACE-III also has good validity compared with the MMSE, in which the diagnosis of dementia has higher accuracy than that of the MMSE.

#### 2.2.6. Use of Digital Devices

The participants were asked if they use any computer or any digital device (mobile phone, tablet, or laptop). Next, the participants answered a question on the number of hours they use the device on average in a week, followed by a question on the purpose of use of the device (communication, social media, entertainment video, games, work, news, books, or others), where participants could tick more than one answer.

### 2.3. Data Analysis

Data analysis was performed using SPSS version 26.0 with the PROCESS 4.1 plug-in. An independent *t*-test was applied to compare cognitive reserve, the risk of cognitive impairment, and cognition between healthy older adults who use digital devices mainly for communication and who use digital devices for multiple purposes. In this analysis, healthy older adults who mainly use digital devices for communication (phone calls and messaging) were categorised into one group, while healthy older adults who use digital devices for multiple purposes, such as using a smartphone and computer (for social media, entertainment video games, reading, work, or purposes other than communication) were categorised in another group. Then, the cognitive reserve, risk of cognitive impairment, and cognition were compared between these two groups using independent t-tests. Next, Pearson’s correlation or Kendall’s tau-b was applied to determine the relationship between cognitive reserve with the risk of cognitive impairment, and the relationship between cognitive reserve with cognition. Following that, a multivariate linear regression analysis was applied to test the prediction of total hours spent on digital device use towards cognitive reserve, the risk of cognitive impairment, and cognition. Lastly, PROCESS 4.1 was used for mediation and moderation analysis [35].

### 2.4. Ethical Considerations

This study was approved by Research Ethics Committee of Universiti Kebangsaan Malaysia (RECUKM). The approval no. is JEP-2021-51. Informed consent was obtained from all the participants before the administration of questionnaires.

## 3. Results

### Participant Characteristics

A total of 256 participants were screened for this study. Out of the 256, 16.4% were excluded from the study. Among the excluded data, 18 were excluded due to MMSE score (<23), 14 were excluded due to M-GDS-14 score (<5), and another 6 were excluded due to both MMSE (<23) and M-GDS-14 (<5) scores. One outlier was removed from the data due to extreme scores in CSR, and three outliers were removed for extreme scores in total hours spent per week on digital device use. In the end, a total of 210 participants were included in the final data analysis. The descriptive data are summarised in Table 1. Participants’ ages range from 60 years old to 82 years old, with a mean of 67.04 (SD = 5.11). The mean number of years of education is 11.5 (SD = 4.79). In summary, 54.9% of the participants are male, 76.9% of the participants are married, 53.8% of the participants are Chinese, the level of job was classified into five levels, and 66.7% of the participants are no longer working.

In order to test hypothesis (a), three independent *t*-tests were conducted, respectively, for cognitive reserve, the risk of cognitive impairment and cognition, between the group using digital devices for communication only (*n* = 14) and with the group of using digital devices for multiple purposes that require more cognitive effort (*n* = 196). All the skewness and kurtosis are in the acceptable range [36]. Thus, preliminary assumption testing indicated normality was not violated for the group using digital devices for communication only or the group using digital devices for multiple purposes. Levene’s test for equality of variances was found to be not significant for cognitive reserve (F = 0.08, *p* > 0.05), the risk of cognitive impairment (F = 1.97, *p* > 0.05), or cognition (F = 0.07, *p* > 0.05).

The independent *t*-test comparing the cognitive reserve between the two groups was significant (t (208) = −3.30, *p* < 0.05, two-tailed, d = 0.91), indicating cognitive reserve for the group using digital devices for multiple purposes (M = 57.11, SD = 11.29) is higher than cognitive reserve for the group using digital devices for communication only (M = 46.79, SD = 11.49). The independent t-test comparing the risk of cognitive impairment between the two groups was not significant (t (208) = −1.621, *p* > 0.05, two-tailed), indicating the risk of cognitive impairment for the group using digital devices for multiple purposes (M = 13.02, SD = 2.45) did not significantly differ from than the risk of cognitive impairment for the group using digital devices for communication only (M = 11.93, SD = 1.98). The independent t-test comparing cognition between both groups was significant (t (208) = −2.73, *p* < 0.05, two-tailed, d = 0.78), indicating the cognition of the group using digital devices for multiple purposes (M = 83.46, SD = 10.01) was significantly higher than than the cognition of the group using digital devices for communication only (M = 75.93, SD = 9.53).

For hypothesis (b), the bivariate correlation between cognitive reserve and cognition was positive and moderate, with r(208) = 0.30 and *p* < 0.001. Kendall’s tau-b was interpreted. Kendall’s tau-b indicated that the correlation between cognitive reserve and cognition was positive: τ = 0.26, *p* < 0.001, two-tailed, *N* = 210. Kendall’s tau-b indicated that the correlation between cognitive reserve and cognition was positive: τ = 0.17, *p* < 0.001, two-tailed, N = 210.

For hypothesis (c), after controlling for age and marital status, the results suggested that digital device use explained 9% of the variance in cognitive reserve (R^2^ of 0.09, withan F(5,204) = 4.20, *p* < 0.05). Digital device use was significantly associated with cognitive reserve (B = 0.09, t = 2.29, *p* < 0.05). With gender as a control variable, linear regression analysis showed that digital device use explained 5% of the variance in cognitive reserve (R^2^ of 0.05, with an F(2,207) = 5.28, *p* < 0.05). However, further analysis revealed that digital device use was not significantly associated with the risk of cognitive impairment (B = 0.02, t = 1.97, *p* = 0.05).

To test hypothesis (d), as digital device use did not predict the risk of cognitive impairment, a series of mediating and moderating analyses were conducted for the mediating and moderating effect of cognitive reserve in the relationship between digital device use and cognition. The total model is illustrated in Figure 1 and the mediation model is illustrated in Figure 2.

As displayed in Figure 1, the first step of the mediation model, the total effects showed that the total hours per week spent on digital device use with cognition was significant (β = 0.17, t(208) = 4.98, *p* < 0.001). The following analysis, as indicated in Figure 2, showed that the direct effect of the total hours spent on digital device use per week on cognitive reserve was also significant (β = 0.13, t(208) = 3.21, *p* < 0.001). Next, the direct effect analysis showed that cognitive reserve significantly affects cognition (β = 0.29, t(208) = 5.18, *p* < 0.001). Finally, the analysis of the indirect effects revealed that cognitive reserve significantly mediates the relationship between total hours per week spent on digital device use and cognition, with β = 0.04 and *p* < 0.01 (95% CI, 0.012 to 0.066). Cognitive reserve accounted for 21.8% of the total effect. The results of the mediation analysis are summarised in Table 2 and Table 3.

Meanwhile, the overall model results indicated that the moderating effects of cognitive reserve on the relationship between the total number of hours per week spent on digital device use and cognition was significant (F(3,206) = 18.69, *p* < 0.001, R^2^ = 0.21). Individually, the total number of hours per week spent on digital device use did not significantly predict cognition (β = 0.30, t(206) = 1.93, *p* > 0.05). On the other hand, one unit of cognitive reserve significantly predicted a 0.36 unit increase in cognition (β = 0.36, t(206) = 4.17, *p* < 0.001). However, cognitive reserve did not significantly moderate the relationship between the total number of hours per week spent on digital device use and cognitive reserve (β = −0.003 t(206) = −1.13, *p* > 0.05). The results are summarised in Table 4 and Table 5. Meanwhile, Figure 3 illustrates the moderation model for the moderation analysis.

## 4. Discussion

Among different cognitive stimulating activities that contribute to cognitive reserve, digital device use has good potential to be utilised by healthy older adults as it is easily accessible and portable. This study provides a new perspective on the usage of digital devices among healthy adults because it was discovered that there was a large number of older adults who use digital devices for multiple purposes compared to using them for communication only. This relatively new finding contradicts a previous study stating that older adults rarely use digital devices for multiple purposes, such as entertainment, reading, and accessing social media [37,38,39].

There are several possible explanations for this finding. One of the possibilities is that the older adults are recruited from more urban areas in Klang Valley, and this is not representative of older adults who live in rural areas, as older adults living in other states in Malaysia may demonstrate different sociodemographic characteristics and lifestyles [40]. Another possibility is the increasing adoption of digital device use among older adults, where digital devices emerge as a new trend precipitated by the pandemic as the demand increases [11]. For example, in Canada, a survey with 1923 older adults aged 65 years old during a ‘stay at home’ order due to the COVID-19 pandemic indicates older adults reported feeling more isolation, positive feelings about technology, and increased technology use for better health and psychological wellbeing [41]. Given that older adults feel positive about technology and utilised digital devices for various purposes, learning to use digital devices has the potential to further build on cognitive reserve through stimulating the brain network. Learning to use digital devices stimulates the brain through engaging and employing various cognitive strategies to deal with the demands of learning [42]. Thus, using digital devices for multiple purposes is a good sign for healthy older adults, as it was found to be associated with better cognitive reserve and cognition in this study.

Based on the framework of Franzmeier et al. [24], the three constructs that were investigated in this research were composed of cognitive reserve, the risk of cognitive impairment, and cognition. Generally, significant relationships were found between cognitive reserve, the risk of cognitive impairment, and cognition. Specifically, higher cognitive reserve was found to be significantly associated with higher cognition. This result supported the theory of cognitive reserve, in which cognitive reserve has a better reservoir in terms of its cognitive capacity to allow older adults to maintain their cognition [14]. However, this does not mean that having a higher cognitive reserve would protect against pathology. For instance, a previous study suggested that older adults who have higher cognitive reserve were associated with better cognition, but there was no effect found in the rate of cognitive deterioration among older adults with mild cognitive impairment [23]. Based on this argument, it is too early to conclude that cognitive reserve protects against the progression of pathology. Remarkably, this study found that higher cognitive reserve was associated with a lower risk of cognitive impairment. With that being said, although cognitive reserve may not provide ultimate protection against cognitive impairment when pathology is significant, the role of cognitive reserve is still important among healthy older adults as it offers a buffer against the risk of cognitive impairment. Therefore, cognitive reserve attenuates the risk of cognitive impairment and links to better cognition among healthy older adults [43,44].

This study is interested in finding out whether the use of digital devices can be associate with cognitive reserve, the risk of cognitive impairment, and cognition. As hypothesised, digital device use was positively associated with cognitive reserve and cognition, as expected. On the other hand, digital device use was not found to be significantly associated with the risk of cognitive impairment. That indicates that although digital device use can support cognitive reserve and cognition through building on cognitive resources and capacity, it does not have an effect on changing other risk factors of cognitive impairment, as the risk for cognitive pathology remains the same for healthy older adults. In order to enumerate this notion, the underlying mechanism of digital device use towards cognitive reserve is different from the underlying mechanism with the risk of cognitive impairment, in which healthy older adults benefit from digital device use in terms of cognitive reserve and cognition regardless of their risk of cognitive impairment. Henceforth, the longer healthy older adults use digital devices, the higher their cognitive reserve and cognition. It is important to note that intervention would be more effective at an early stage, especially before the onset of cognitive impairment [44]. As such, digital devices emerge as an early intervention for building cognitive reserve.

In previous literature, cognitive reserve was found to have a mediating and moderating effect on cognition [25,27]. However, there was a lack of studies that examine this relationship with digital device use. The findings from this study suggested that cognitive reserve partially mediated the relationship between digital device use and cognition. In this case, this finding supports the notion that, at least partially, digital device use can lead to better cognition through an underlying mechanism of cognitive reserve. As such, healthy older adults who use digital devices for a longer period of time have better cognition because of cognitive reserve and some other reasons. Fundamentally, the theory of cognitive reserve asserts that the brain compensates for pathology through applying active strategies within the available capacity [14].

Although there was no strong evidence for a neuronal network of cognitive reserve, previous studies suggested that cognitive reserve involves interrelated cognitive processes such as arousal, sustained attention, response to novelty, attention, executive functioning, and working memory [45,46]. At the same time, the use of digital devices requires learning and applying various cognitive strategies, stimulating cognitive activities in the brain across different domains. This involves sustaining the attention to operate the digital device and receiving information from the digital device in order to manipulate the information for further processing. In particular, digital device use was posited to be associated with cognitive processes such as reasoning, working memory, and processing speed [47]. Given that there are some overlaps in terms of the cognitive mechanism underlying digital device use and cognitive reserve, this provides a possible explanation of how cognitive reserve mediates the relationship between digital device use and cognition. In essence, the cognitive stimulation enkindled by digital device use requires flexible cognitive strategies that build on the reservoir of cognitive reserve, leading to better cognition as a result. Therefore, cognitive reserve plays a partially mediating role as digital device use manifests its benefits through the underlying mechanism of cognitive reserve.

Nonetheless, this study’s findings did not support the hypothesis that cognitive reserve has a moderation effect on the relationship between digital device use and cognition, despite a significant mediation effect. The finding of an insignificant moderation effect contradicted a previous study in which Jokinen et al. [27] found cognitive reserve to moderate the relationship between cognitive pathology and cognition. Thus, the finding of this study postulated that even though cognitive reserve mitigates the effect of cognitive impairment on cognition, cognitive reserve does not enhance or change the relationship between digital device use and cognition. Comparatively, previous studies that found a significant moderation effect of cognitive reserve on cognition were conducted among older adults who have cognitive impairment, whereas this study was conducted among healthy older adults [27,45]. Thus, cognitive reserve may operate according to a different mechanism among healthy older adults compared with older adults with cognitive impairment. In this case, this study filled the gap of knowledge where no moderation effect of cognitive reserve was found between digital device use and cognition. Therefore, the current study extends the knowledge in understanding of the role of cognitive reserve in the enhancement of cognition among healthy older adults.

### 4.1. Limitations

Due to the sampling method in this study, in which recruitment was carried out in the community and most of the participants are healthy older adults, there is a greater chance of them having greater involvement in social activities and a greater need to use digital devices [48]. In the sample of this study, most of the participants use digital devices for multiple purposes. In this case, we caution against making generalisations to other populations who are more isolated and have greater health risks. Apart from that, the sample size of 210 participants is relatively small and from a single urban area, limiting the ability to generalise findings to the larger older adult population in Malaysia. Also, the recruitment process through convenience sampling with strict exclusion criteria may cause a biased sample that limits the generalisability of this study. In addition to that, another methodological consideration for this study is the length of the questionnaires. The administration of the questionnaire involves cognitive assessments such as M-MMSE and ACE-III. Considering that the participants are older adults, sustained attention is expected to decline during old age, and thus the performance might be affected by the prolonged administration of multiple questionnaires [49]. Taking this into consideration, the administration was kept concise and precise.

Besides that, the limitation of a cross-sectional study is that it does not permit the interpretation of cause and effect [39]. Although there was an association found between digital device use and cognitive reserve and cognition, the conclusion that digital devices cause higher cognitive reserve and cognition cannot be made. Vice versa, the effect can be a reverse causality in which individuals with higher cognitive reserve and cognition tend to use digital devices more as they have a larger cognitive capacity. However, Spector [50] argued that other study designs such as longitudinal and prospective designs would prove cause and effect. This is due to the failure in selecting time points in the research. Despite the fact that a causal relationship is hard to establish in a cross-sectional study, this study’s strength is its exploratory nature to explain cognitive reserve with the risk of impairment and cognition. In addition to that, control variables were added to rule out alternative explanations. Furthermore, a cross-sectional study offers the opportunity to examine the effects of using digital devices in everyday life, which can be a naturally occurring phenomenon among today’s older adults compared to interventions that are well-designed by practitioners. Although there is limitation in determining causal relationships between digital device use, cognitive reserve, and cognition, this does not denigrate the explanatory power of a cross-sectional study that was well conducted with the inclusion of control variables.

### 4.2. Future Research Direction

The findings from this study act as a catalyst for future research to identify whether the continuous use of digital devices can promote better benefits for cognition among older adults in the longer term. It was assumed that during the COVID-19 epidemic and onwards, a large number of older adults started to use digital devices [11]. Looking into the future, it is foreseen that older adults will depend more and more on digital device use. In the long run, there is potential to conduct longitudinal studies to examine the effect of long-term digital device use on cognitive reserve and cognition. In addition, future research can also look into identifying the neural networks that relate digital device usage to cognitive reserve [51]. As the mediation effect of cognitive reserve between digital device use and cognition was found to be partial, future research can also identify the other factors that mediate this relationship. Digital device use addiction is a rapidly growing concern among older adults [52]. Therefore, future researchers need to consider how the over-use of phones may lead to negative consequences for healthy brains. In summary, this study provides some insight into future directions for research on digital device use and different constructs of brain function among older adults.

### 4.3. Implications

In terms of implications, this study’s findings suggest that the daily use of digital devices helps with older adults’ cognitive reserve and cognition. Thus, older adults can be encouraged to reap the benefits of digital device use. Although this study showed that there are more older adults using advanced digital devices, e.g., smartphones, a number of older adults do not use any digital devices or have no access to digital devices. In order to increase digital device uses among older adults, social influence is one of the possible interventions to invigorate the use of digital devices among older adults [53]. However, there are barriers for society to influence older adults to use digital devices. One of the barriers is the misconception that older adults do not use digital devices, although most older adults already utilise digital devices [54]. Henceforth, the community, including practitioners and family members, should curtail their stereotypes and bias towards older adults using digital devices and try to influence their older adults to utilise digital devices in order to reap their benefits. Nevertheless, it is important not to overlook that digital device use cannot predict the risk of cognitive impairment. With that being said, it is impractical to rely on digital device use alone to mitigate the effect of cognitive deterioration. As the effect of digital device on cognition is explained by cognitive reserve, which acts as the underlying mechanism, building on cognitive reserve through other activities is also important.

## 5. Conclusions

In conclusion, this study’s findings suggest that the daily use of digital devices helps with cognitive reserve and cognition in older adults. Thus, older adults are encouraged to reap the benefits of digital device use. Although this study showed that there are a greater number of older adults using more advanced digital devices now, there are still a number of older adults that do not use a digital device, despite the fact that there is potential to benefit from their use. Although digital device use is not associated with the risk of cognitive impairment, it is impractical to rely solely on digital devices to mitigate the effect of cognitive deterioration. As the effect of digital devices towards cognition is explained by cognitive reserve, which acts as the underlying mechanism, building on cognitive reserve through other activities is also important.

## Figures and Tables

**Figure 1 healthcare-11-02822-f001:**
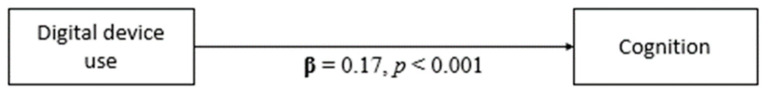
Total effect model between digital device use and cognition.

**Figure 2 healthcare-11-02822-f002:**
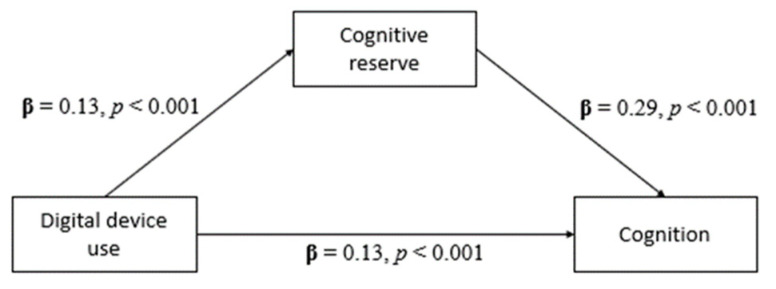
Mediation model of cognitive reserve mediating the relationship between digital device use and cognition.

**Figure 3 healthcare-11-02822-f003:**
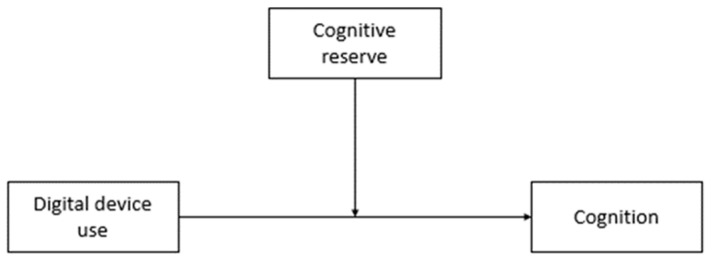
Moderation model of cognitive reserve moderates the relationship between digital device use and cognition.

**Table 1 healthcare-11-02822-t001:** Descriptive statistics.

Variable	Total Sample (*N* = 210)		Skewness		Kurtosis	
	M(SD) or *n*(%)	Range	Stat.	SE	Stat.	SE
Age (years), M(SD)	67.22 (5.24)	60–83				
Gender, *n*(%)						
Male	114 (54.3)					
Female	96 (45.7)					
Marital Status, *n*(%)						
Single	16 (7.6)					
Married	161 (76.7)					
Divorced	5 (2.4)					
Widowed	28 (13.3)					
Ethnicity, *n*(%)						
Malay	68 (32.4)					
Chinese	113 (53.8)					
Indian	25 (11.9)					
Others	4 (1.9)					
Education level, *n*(%)						
No formal education	4 (1.9)					
Primary school	39 (18.6)					
Secondary school	96 (45.7)					
Diploma/certificate equivalent	31 (14.8)					
Undergraduate	16 (7.6)					
Graduate and above	24 (11.4)					
Currently working, *n*(%)						
Yes	66 (31.4)					
No	144 (68.6)					
Level of job, *n*(%)						
Low skilled manual work	47 (22.4)					
Skilled manual work	35 (16.7)					
Skilled non-manual work	54 (25.7)					
Professional occupation	56 (26.7)					
Highly responsible or intellectual occupation	18 (8.6)					
Digital device ownership, *n*(%)						
Mobile phone only	5 (2.4)					
Smartphone only	108 (51.4)					
Two devices and above including tablet or laptop	97 (46.2)					
Purpose of digital device use, *n*(%)						
Communication only	14 (6.7)					
Multipurpose	196 (93.3)					
Total hours per week on digital device use, M(SD)	26.51 (20.02)	1–89	0.81	0.17	0.20	0.33
M-CSR, M(SD)	56.42 (11.57)	23–81	−0.20	0.17	−0.55	0.33
TUA-WELLNESS, M(SD)	12.94 (2.43)	6–17	−0.50	0.17	−0.05	0.33
ACE-III, M(SD)	83.96 (10.14)	45–99	−0.99	0.17	0.89	0.33

**Table 2 healthcare-11-02822-t002:** Mediation estimates.

Total Effects			Direct Effect				Indirect Effects			
β	*t*	*p*-Value	β	*t*	*p*-Value	Hypothesis	β	*SE*	Percentile Bootstrap 95% Confidence Interval	
									Lower	Upper
0.17	4.98	<0.001 **	0.13	4.03	<0.001 **	H:DDU > CR > C	0.04	0.14	0.012	0.066

Note: Significant at ** *p* < 0.001.

**Table 3 healthcare-11-02822-t003:** Summarised path analysis.

	*R*	*R^2^*	*F*	β	*SE*	*t*	*p*-Value
Digital device use → Cognition (Path c)	0.33	0.11	24.81	0.17	0.20	4.98	<0.001 **
Digital device use → Cognitive reserve (Path a)	0.22	0.05	10.30	0.13	0.04	3.21	0.002 *
Cognitive reserve → Cognition (Path b)	0.46	0.21	27.37	0.29	0.56	5.18	<0.001 **
Digital device use lessen predicts cognitive reserve (Path c’)	0.46	0.21	27.37	0.13	0.03	4.03	<0.001 **

Note: Significant at * *p* < 0.05; ** *p* < 0.001.

**Table 4 healthcare-11-02822-t004:** Moderation model summary.

Moderator	*R*	*R* ^2^	*F*	*p*-Value
Cognitive reserve	0.46	0.21	18.69	<0.001 **

Note: Significant at ** *p* < 0.001.

**Table 5 healthcare-11-02822-t005:** Moderation estimates.

Variables	β	*SE*	*t*	*p*-Value
Digital device use	0.300	0.155	1.93	0.054
Cognitive reserve	0.363	0.087	4.17	<0.001 **
Digital device use * Cognitive reserve	−0.003	0.003	−1.13	0.262

Note: Significant at * *p* < 0.05; ** *p* < 0.001.

## Data Availability

Data can be attained from the corresponding author.

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
