# Peer review of "Digital Device Use, Risk of Cognitive Impairment, and Cognition in Healthy Older Adults: The Role of Cognitive Reserve"

_healthcare, 2023, doi:10.3390/healthcare11212822_

Round 1

Reviewer 1 Report

This study aimed to investigate the relationship between digital device use, cognitive reserve, risk of cognitive impairment, and cognition among healthy older adults. A total of 210 participants were recruited for a cross-sectional study. The results revealed that there was a significant difference in cognitive reserve and cognition between older adults who used digital devices for communication purposes only and those who used them for multiple purposes. A significant relationship was also found between cognitive reserve, risk of cognitive impairment, and cognition. While digital device use predicted cognitive reserve and cognition, it did not significantly predict the risk of cognitive impairment. Cognitive reserve was found to partially mediate the relationship between digital device use and cognition, suggesting that it plays a role in this association. The study suggests that digital device use can be a beneficial daily intervention for healthy older adults to enhance their cognitive reserve and potentially protect their cognition from decline. However, it also emphasizes the importance of exploring additional activities to further enhance cognitive reserve in this population.

Introduction section

The introduction provides a general overview of the ageing population issue, but it lacks a clear and concise research objective or research question to guide the study. While the importance of preparing for the ageing population is mentioned, the specific relevance and significance of investigating the impact of digital device use on cognitive function in older adults is not clearly stated. The literature review on successful ageing and neuroprotective factors seems disconnected from the main topic of digital device use and cognitive reserve. In addition, the transition between discussing digital device use and cognitive reserve is abrupt and lacks a clear explanation of how these concepts are related. The hypotheses proposed at the end of the introduction are not well integrated into the preceding discussion and would benefit from a better contextualization within the study's objectives.

Methods section

Firstly, the sample size of 210 participants appears to be relatively small for a quantitative cross-sectional study. A larger sample size would have provided more robust results and increased the generalizability of the findings.

Secondly, the recruitment process through purposive sampling may introduce bias into the study. It is unclear how participants were selected and whether they represent a diverse range of older adults in terms of socio-economic background, education level, and health status. This lack of diversity could limit the external validity of the study and the generalizability of the findings to a broader population of older adults.

Furthermore, the exclusion criteria seem to be quite restrictive and may have led to a biased sample. Excluding individuals who do not use digital devices, have cognitive or psychiatric problems, or have serious chronic illnesses, including dementia, heart disease, and cancer, could significantly limit the representation of older adults with varying levels of cognitive function and health conditions. Consequently, the findings may not be applicable to the broader population of older adults.

Lastly, it would be helpful to have more details about the sociodemographic characteristics of the participants, such as their age range, gender distribution, educational background, and socioeconomic status. These factors can significantly influence cognitive reserve and the use of digital devices, and their inclusion would strengthen the study's analysis and interpretation.

Discussion section

While the study provides interesting insights into the potential benefits of digital device use among healthy older adults for cognitive reserve and cognition, there are some points that require further consideration.

The study highlights the relatively high usage of digital devices for multipurpose activities among older adults, which contradicts previous studies suggesting that older adults primarily use digital devices for communication purposes only. However, the authors do not thoroughly discuss the reasons for this discrepancy. While they mention possible explanations, such as the urban area recruitment or the increasing adoption of digital devices due to the pandemic, further exploration and analysis of these factors would strengthen the discussion and enhance the understanding of the findings.

Furthermore, the study suggests a significant relationship between cognitive reserve, risk of cognitive impairment, and cognition, which aligns with the theory of cognitive reserve. However, the authors could have provided more in-depth analysis and discussion of these relationships, particularly in relation to previous research. The implications of these findings and their relevance for understanding cognitive function among healthy older adults should be elaborated upon.

In terms of the predictive value of digital device use, the authors indicate that it significantly predicts cognitive reserve and cognition but does not predict the risk of cognitive impairment. However, this conclusion lacks sufficient explanation or interpretation. It would be beneficial for the authors to discuss the potential reasons why digital device use may have differential effects on these cognitive outcomes and provide a more comprehensive analysis of these findings.

Moreover, the discussion of the mediation and moderation analyses is somewhat limited. While the study identifies a partial mediation effect of cognitive reserve on the relationship between digital device use and cognition, the authors do not delve into the underlying mechanisms or provide a detailed explanation of this finding. Additionally, the lack of a moderation effect of cognitive reserve on the relationship between digital device use and cognition is briefly mentioned but not thoroughly explored. Further discussion and interpretation of these results would enhance the understanding of the role of cognitive reserve in the context of digital device use among healthy older adults.

The section on limitations and future research directions is relatively concise and could benefit from further elaboration. The limitations of the study, such as the sampling method and the administration of lengthy questionnaires, are mentioned briefly, but their potential impact on the study's findings and implications should be discussed more comprehensively. Additionally, while the authors suggest future research directions, such as longitudinal studies and investigating the neural networks involved in digital device use and cognitive reserve, these ideas could be expanded upon to provide a more comprehensive roadmap for future investigations.

Author Response

Dear Reviewer,

Thank you

Reviewer 2 Report

Interesting study but with a major issue. Authors conducted a cross-sectional quantitative study what does not allow conclusions on causal relationships but only associations. However, they define one of study goals as “Digital device use among healthy older adults will significantly predict cognitive reserve, risk of cognitive impairment and cognition among healthy older adults.” To predict something means that there is a causal relationship between two factors. This is not the case here.

L24-25 “Although digital device use was found to significantly predict

cognitive reserve and cognition, it did not significantly predict the risk of cognitive impairment…”  I think, indeed, digital device use was found to be significantly associated with a cognitive reserve and cognition, but not with the risk of cognitive impairment.” Please check all manuscript in terms of using appropriate epidemiological terms.

L. 202-204 „This section may be divided by subheadings. It should provide a concise and precise

description of the experimental results, their interpretation, as well as the experimental

conclusions that can be drawn.”  This text is not needed and should be removed

L 247 correlation of 0.26 cannot be considered strong. for absolute values of r, 0-0.19 is regarded as very weak, 0.2-0.39 as weak, 0.40-0.59 as moderate, 0.6-0.79 as strong and 0.8-1 as very strong correlation. Please also check other correlation coefficients in terms of interpretation.

L.293 “Table Error! No text of specified style in document.”

Author Response

Dear Reviewer,

Thank you

Reviewer 3 Report

The neuroprotective factor is essential as a part of successful ageing. As such, digital device use was proposed to be an easily accessible and available cognitive stimulating activity to enhance brain function. Nonetheless, there were lack of studies in inspecting the connection between digital device use with cognitive reserve, risk of cognitive impairment, and cognition.

The authors aim to investigate the potential mediator and moderator of the association between digital device use, cognitive reserve, risk of cognitive impairment, and cognition among healthy older adults.

They conducted a quantitative cross-sectional to investigate the relationship between digital device use with cognitive reserve, risk of cognitive impairment, and cognition. A total of 210 healthy older adults recruited through purposive sampling.

Their results  revealed that there was a significant difference in cognitive reserve and cognition between healthy older adults who use the digital device for communication purpose only and who use the digital device for multipurpose. A significant relationship was also found between cognitive reserve, risk of cognitive impairment, and cognition. Although digital device use was found to significantly predict cognitive reserve and cognition, it did not significantly predict the risk of cognitive impairment. Cognitive reserve partially mediated the relationship between digital device use and cognition, supporting the notion that cognitive reserve act as an underlying mechanism in the relationship between digital device use and cognition.

 Hence, digital device use was suggested to be a good daily intervention for healthy older adults to build on their cognitive reserve and potentially protect their cognition from declining. Nevertheless, relying on digital device use alone is not sufficient and other activities should be explored to enhance cognitive reserve among healthy older adults.

There is a strong need of these studies facing the cognitive stimulation of brain by means of the digital devices.

I have some minor suggestions for the authors:

1.       Use “ [  ]” to cite references.

2.       Add the aim after the hypotheses.

3.       Par. 2.2 should be improved. It currently appear as a list. Please introduce the parts by means of some sentences and arrange into subparagraphs according to the MDPi standards.

4.       The figures (for example the model in figure 1 and 2) must be described in the body of the manuscript in details.

5.       “Table Error! No text of specified style in document. Moderation model summary” Correct this.

6.       Generally you must follow the MDPI standard in the body of the manuscript.

Author Response

Dear Reviewer,

Thank you

Reviewer 4 Report

The cross-sectional study shows a good effort in trying to investigate the potential mediator and moderator of the association between digital device use, cognitive reserve, risk of cognitive impairment, and cognition among healthy older adults.

The manuscript is well written and organized. Only editing of English language should be done.

Minor editing of English language required.

Author Response

Dear Reviewer,

Thank you

Reviewer 5 Report

The present paper present pertinent data regarding the neuroprotective role of good use of electronic devices on elderly. The methods and data presented in the manuscript are clear and provide considerable intervention in the understanding of the implication of phone -use in preventing cognitive decline at old ages. 

Nevertheless, this work could be improved throughout considering some editing recommendations. Here are some that I find most pertinent:

In the introduction: 

- The first paragraph: In my opinion, aging population is not the real issue that have to be denoted for this study. It is instead the health issue of aging population that have to be considered.

- Please reformulate the third paragraph. The term " the neuroprotective factor" is misunderstood, do you mean by it " Neuroprotective factors" instead? 

- In the last paragraph, please explain how does it come that cognitive decline is not affected by cognitive reserve? 

Please consider mentioning the spelled-out version of abbreviations before acronyms at every first-time mentioned in the text. 

Regarding reliability of the study, isn't it provided in the literature that overuse of phones may lead to addiction? How may this contradict to your finding?

The quality of writing sounds good. Nevertheless, authors may consider minor English language editing for the manuscript. Especially the introduction, it could be improved by using clearer expressions.

Author Response

Dear Reviewer,

Thank you

Round 2

Reviewer 1 Report

have no more comment and would like to thank the authors for this sufficient revision. I recommending to accept this paper. 

Reviewer 2 Report

no comments